# Learning Precise Mask Representation for Siamese Visual Tracking

**DOI:** 10.3390/s25185743

**Published:** 2025-09-15

**Authors:** Peng Yang, Fen Hu, Qinghui Wang, Lei Dou

**Affiliations:** The National Key Laboratory of Transient Physics, School of Nanjing University of Science and Technology, Nanjing 210094, China; pyang_15@163.com (P.Y.);

**Keywords:** visual object tracking, siamese network, deep learning, segmentation mask, saliency localization

## Abstract

Siamese network trackers are a prominent paradigm in visual object tracking due to efficient similarity learning. However, most Siamese trackers are restricted to the bounding box tracking format, which often fails to accurately describe the appearance of non-rigid targets with complex deformations. Additionally, since the bounding box frequently includes excessive background pixels, trackers are sensitive to similar distractors. To address these issues, we propose a novel segmentation-assisted model that learns binary mask representations of targets. This model is generic and can be seamlessly integrated into various Siamese frameworks, enabling pixel-wise segmentation tracking instead of the suboptimal bounding box tracking. Specifically, our model features two core components: (i) a multi-stage precise mask representation module composed of cascaded U-Net decoders, designed to predict segmentation masks of targets, and (ii) a saliency localization head based on the Euclidean model, which extracts spatial position constraints to boost the decoder’s discriminative capability. Extensive experiments on five tracking benchmarks demonstrate that our method effectively improves the performance of both anchor-based and anchor-free Siamese trackers. Notably, on GOT-10k, our method increases the AO scores of the baseline trackers SiamRPN++ (anchor-based) and SiamBAN (anchor-free) by 5.2% and 7.5%, respectively while maintaining speeds exceeding 60 FPS.

## 1. Introduction

Visual object tracking is a challenging computer vision task. Given the initial target state in the first frame, the goal is to precisely determine and report the target extent in the subsequent video sequence. Currently, the popular tracking paradigm is correlated bounding box tracking [1,2], where targets represented by axis-aligned rectangles are regressed by cross-correlation between the template and search patches. This paradigm performs well with low-dimensional transformations, e.g., rigid deformation or translation, but struggles with more complex scenarios, such as those inducing irregular deformation due to occlusion or aspect ratio variation. In these cases, the bounding box only reports a rough target extent in the image, often including significant background pixels with limited target information, rendering it unreliable.

Recently, Siamese network trackers have gained popularity as a leading bounding box-based approach, primarily due to their powerful similarity learning capability. Building on this Siamese structure [3,4], numerous works have focused on predicting high-quality bounding boxes, such as approximate exhaustive anchor-based [5,6,7,8,9] and keypoint-based anchor-free [10,11,12,13,14,15] methods. While these improve target representation by generating more compact axis-aligned bounding boxes, they fare poorly in tracking non-rigidly deforming objects misaligned with the image axis. For instance, when objects are rotated, elongated, or stretched within the plane, even cutting-edge bounding box-based trackers like SiamBAN [12] and SiamRPN++ [7] often fail to report the target accurately and start tracking distractors (see Figure 1 left subplot). In such cases, the binary mask serves as the ideal target representation model, accurately delineating the target extent in the image on a pixel-by-pixel basis.

Moreover, some works have successfully revealed the superiority of segmentation masks in characterizing target states [16,17,18,19,20,21]. Among them, a typical approach is bounding-box-centric [16], e.g., Siam R-CNN [17] and Alpha-Refine [18], where final segmentation masks are refined from predicted bounding boxes using additional independent post-processing mask networks. Although these methods achieve pixel-level target representation, the mask network merely supplements the bounding box prediction module, serving as an auxiliary enhancement rather than significantly improving the localization accuracy. Conversely, a segmentation-centric paradigm, inspired by video object segmentation (VOS) [22], focuses on directly using segmentation masks to represent targets without estimating bounding boxes beforehand. Given that VOS typically handles large target objects in short-term scenarios with fewer distractors, it is unsuitable for tracking small targets in cluttered background scenes. D3S [20] thus utilizes correlation filters to extract positional information and video matching [23] to obtain foreground similarity and posterior probability. These supplementary details about pixel-level target existence are then integrated into the mask refinement module to alleviate the sensitivity to distractors. RTS [16] designs an instance localization branch for precise recognition and localization. It also constructs separate segmentation memory and instance memory components to store masks and localization features predicted from previous frames, which are then used to guide the update of decoders. Compared to Siamese network trackers, these segmentation trackers are significantly more complex, resulting in a substantial speed reduction. Additionally, these dedicated segmentation components are difficult to transfer to other tracking algorithms, limiting their general applicability.

To achieve simple and efficient segmentation tracking, this paper proposes a precision mask representation (PMR) module, which is easily plugged into various bounding box-based Siamese trackers to enable accurate pixel-wise target tracking. In detail, the proposed module serves as a mask network parallel to the baseline pipeline rather than a sequential addition following the classification and regression branches. Our component features a U-shape encoder-decoder architecture, where encoders extract saliency from multilayer backbone features and decoders process this saliency to generate binary masks with multiple receptive fields. Unlike prevailing mask generation networks [20,21] that predict masks from final fusion features, our module performs multi-stage mask predictions, comprehensively learning the saliency of each layer feature. Additionally, we design a saliency localization (SL) head to further enhance the segmentation robustness of the PMR module. This head focuses the module on the saliency feature of the target object, thereby eliminating the segmentation bias in PMR. We integrate the proposed PMR module and SL head into typical anchor-based and anchor-free Siamese trackers, such as SiamRPN++ [7] and SiamBAN [24], resulting in improved versions named SiamRPN++-PRMSL and SiamBAN-PMRSL. Extensive experiments on five tracking benchmarks demonstrate the effectiveness of our approach. On GOT-10k [25], the improved trackers achieve related gains of 5.2% and 7.5% over the baseline, respectively (see Figure 1, right subplot). Importantly, the improved trackers maintain speeds above 60 FPS on an RTX 4070 GPU.

Overall, the contributions of our work can be summarized as follows:A simple and efficient precise mask representation module is proposed to address limitations of the bounding box-based tracking paradigm in accurately estimating target extent, enabling pixel-wise segmentation by auxiliary learning multi-scale masks of the target.To enhance the discrimination of our module, a saliency localization head is designed to capture the spatial saliency of targets and suppress similar distractors.The developed PMR module and SL head are generic and easy to integrate into Siamese frameworks, allowing these trackers to achieve accurate segmentation tracking and improve performance without significant additional cost.

## 2. Related Work

### 2.1. Siamese Visual Tracking

Visual object tracking has been dominated by various paradigms over the years. A significant and influential line of work is based on Discriminative Correlation Filters (DCFs) [26,27], which are renowned for their high computational efficiency. With the development of deep learning, deep features are introduced into DCFs to enhance performance [28,29]. Meanwhile, another end-to-end tracking paradigm based on deep learning has emerged, with Siamese network tracking gradually becoming the mainstream approach. Siamese trackers [3,4] formulate tracking as a similarity matching paradigm by calculating the cross-correlation between the template and search patches. The matching function is trained exclusively during the offline phase and remains fixed during the inference phase, achieving speeds of up to several hundred FPS. This remarkable efficiency has consequently attracted widespread attention. Recently, numerous optimization strategies have been proposed to improve tracking performance. SiamRPN [5] first introduced an anchor-based region proposal network (RPN) [30] into the Siamese structure to predict bounding boxes with an adaptive aspect ratio. It decomposed tracking into a classification part for predicting positive anchors and a regression part for estimating the offset between anchor and ground truth. C-RNN [6] cascaded multiple RPNs to exploit both deep semantic and shallow spatial information comprehensively and performed multi-step regression and multi-stage anchor adjustments to estimate more accurate bounding boxes. DaSiamRPN [9] addressed the imbalance in training positive and negative sample classes by introducing a data augmentation scheme to improve model discrimination. SiamRPN++ [7] and SiamDW [8] successfully employed deep convolution networks as the backbone, significantly improving accuracy. Despite the remarkable performance of these anchor-based trackers, the heuristic prior strategy of approximate exhaustive search is sensitive to hyperparameters of anchors, such as ratios and numbers. This limitation motivated the rise of anchor-free approaches. For example, some [24,31] drew inspiration from object methods like FOCS [32], directly classifying all feature points in the response map and regressing their bounding boxes. Others [33,34,35] localized targets by predicting key points, i.e., the center, top-left, and bottom-right corners. These anchor-free methods avoid the cumbersome setup of proposal anchors and exhibit powerful generalization. Additionally, Refs. [36,37] tackled the misalignment problem caused by unreasonable sample distribution or weak interaction between classification and regression branches through the integration of a self-supervised strategy. Refs. [38,39] implemented a memory mechanism to learn temporal information, dynamically updating templates to enhance robustness.

However, the aforementioned Siamese trackers are all based on bounding boxes, which only predict compact axis-aligned bounding boxes as much as possible. This representation format is not optimal, especially for tracking non-rigid targets with irregular deformations. This work aims to optimize the representation paradigm of existing Siamese-based trackers to enhance tracking accuracy.

### 2.2. Visual Object Segmentation

Semi-automatic video object segmentation (VOS) [22] aims to segment the target pixel of interest in video frames according to initial annotations. In contrast to generic object tracking, VOS emphasizes a more precise target representation. Early VOS algorithms [40,41] utilized offline-trained networks to learn generic segmentation features and then fine-tuned these networks online to obtain target-specific representations. To avoid excessive manual hyperparameters during this tuning phase, propagation-based methods [42,43] employed optical flow to infer the current mask from previous frames. Although these methods provided compact and end-to-end VOS solutions, segmentation bias tended to accumulate during propagation, particularly in cases of occlusion. Recently, matching-based methods [44,45] trained end-to-end feature-matching networks to compute correlation scores between upcoming frames and the initial frame for pixel segmentation. Additionally, embedding memory networks were developed to explicitly store previously predicted segmentation masks, aiding in learning temporal feature dependencies and enhancing segmentation robustness. However, VOS predominantly addresses large objects in short-term videos with minimal appearance variations and few distractors. Consequently, directly applying VOS methods to long-term and small-object VOT tasks is not advisable.

### 2.3. Segmentation Guidance Tracking

Inspired by VOS, some studies [16,17,18,20,21] have begun to explore predicting binary masks instead of bounding boxes to delineate target extents, demonstrating superior tracking performance. These works are generally divided into two categories. The first is bounding-box-centric, e.g., Siam R-CNN [17] and Alpha-Refine [18]. They refine images within the bounding boxes obtained through classification and regression branches, using a pre-trained box2reg network to generate binary mask representations. This independent post-processing segmentation network relies on the accuracy of the bounding boxes and does not positively contribute to classification and regression branches. The second category is segmentation-centric, e.g., SiamMask [21], D3S [20], and RTS [16], which directly classify the target mask without prior bounding box prediction. Although the latter represents true segmentation tracking, it suffers from complex network architectures and poor real-time performance. In contrast to the methods above, this paper explores a simple and easily portable segmentation mask network suitable for most bounding box-based Siamese trackers.

## 3. Proposed Method

In this section, we detail our proposed tracking paradigm. Section 3.1 provides an overview. Subsequently, we describe the implementation of the core components: the precise mask representation module in Section 3.2 and the saliency localization head in Section 3.3. Finally, we define the supervised training loss in Section 3.4.

### 3.1. Overview

Most Siamese trackers aim to regress the target’s bounding box and thus operate in this format during the training and inference phases. However, the bounding box inevitably includes background pixels, especially when describing objects that have complex shapes and are not aligned with the image axis. If these background pixels resemble similar objects, the model’s discrimination may be severely degraded. In contrast, the pixel-level binary mask can perfectly segment the target extent, avoiding pixel interference. Thus, this work is dedicated to developing a universal mask plugin that upgrades the representation format without altering the basic architecture of the fully convolutional Siamese network.

The pipeline of our proposed network is illustrated in Figure 2, consisting of four components: the feature extraction backbone, the classification and regression heads, the saliency localization (SL) head, and the precise mask representation (PMR) module. The first two components constitute the foundational elements of the baseline Siamese tracker. The backbone extracts multi-scale features from the input template and search images, followed by an adjustment layer comprising convolution and batch normalization to resize features. For simplicity, we denote the adjusted template and search feature sets as $F^{1,2,3,4,5}=\left\{F\left(1\right),F\left(2\right),F\left(3\right),F\left(4\right),F\left(5\right)\right\}$ and $T^{1,2,3,4,5}=\left\{T\left(1\right),T\left(2\right),T\left(3\right),T\left(4\right),T\left(5\right)\right\}$, respectively. Then, classification and regression heads regress the initial bounding box from the correlation response between $F^{3,4,5}$ and $T^{3,4,5}$.

The latter two components are portable plugins proposed in this paper. The SL head further constrains spatial positional relations between the adjusted template and search features. Specifically, it first uses the Euclidean model to compute distance channels, denoted as $\omega^{3,4,5}=\left\{\Psi \left(3\right),\Psi \left(4\right),\Psi \left(5\right)\right\}$, from the position of the maximum in the response map to all other pixels. Next, $\omega^{3,4,5}$ are multiplied pixel-wise (⊙) with search features to construct saliency features $S^{3,4,5}=\left\{S\left(3\right),S\left(4\right),S\left(5\right)\right\}$, i.e.,(1)S(i)=Ψ(i)⊙F(i),i∈(3,4,5).

The PMR module receives the shallow search features $F^{1,2}$ and the saliency features $S^{3,4,5}$ and then performs multi-stage decoding to obtain multi-field masks. These masks are finally fused to generate a precise target mask.

### 3.2. Precise Mask Representation Module

Recent works have validated the advantages of mask representation for tracking and proposed several mask refinement models. Most of these models depend on a single pathway with stacked convolution and upsampling layers and predict masks from final fusion features. However, this method often results in the degradation of target details due to the smoothing nature of upsampling [46]. To address this, we propose a novel PMR module that adopts a stage-by-stage approach to prevent the feature degradation caused by sequential upsampling. As illustrated in Figure 3, the input shallow and saliency features, i.e., $F^{1,2}$ and $S^{3,4,5}$, first undergo dimension reduction via a 1×1 convolution layer followed by ReLU activation. The adjusted features are then fed into multi-stage mask decoders. To increase the receptive field, the output from each previous stage decoder is concatenated with the input of the current stage decoder. Afterward, they pass through a 3×3 convolution and an upsampling operation to adjust the size to match that of the search image. Finally, these outputs are concatenated and processed through a 1×1 convolution layer followed by a sigmoid function to generate the target mask.

The mask decoder, the main component of the PMR module, identifies whether feature pixels belong to the target. Building on the achievements of previous work on U2-Net [47] in instance segmentation tasks, this paper adapts the residual U-block (RSU) [47] network as the mask decoder (see Figure 4a). Compared to the common single-stream block with direct upsampling and convolution, RSU features a U-shape symmetric encoder–decoder structure to capture local and global contextual information [47]. In detail, an input convolution block, consisting of a convolution layer (3×3 kernel and 1×1 stride), a batch normalization layer, and a ReLU activation, extracts target information f(x) from input features *x*. The encoder side then captures multi-scale information from f(x) through progressive convolutions and pooling operations. These features are then decoded into maps u(f(x)) with local information using progressive upsampling and convolution. Finally, goal and local features are fused by summation, represented as(2)f(x)+u(f(x)).

Note that the height L of the mask decoder corresponds to the number of cascaded encoding convolutions. A larger L results in a deeper RSU block with more pooling operations and multi-scale features with a larger range of receptive fields. According to the spatial resolution of input features, i.e., $F^{1,2}$ and $S^{3,4,5}$, we set the heights L of RSUs in the five mask decoders to 6, 5, 4, 4, and 4, respectively. Furthermore, considering the relatively low resolution of $S^{3,4,5}$ (31×31), further downsampling would lead to a loss of context detail. Therefore, in mask decoders 3 to 5, we employ the variant RSU-4F, which uses dilated convolution instead of pooling and upsampling, as shown in Figure 4b.

### 3.3. Saliency Localization Head

Decoding masks directly from backbone features without discrimination may lead to segmentation mask biases, especially when similar distractors are present near the target. We thus design a saliency localization (SL) head to further impose spatial positional constraints between template and search features and construct discriminative saliency features to mitigate mask biases.

The SL head is derived from the regression head without additional networks; thus, it incurs almost negligible training costs. In the response map of the regression branch, the maximum point is considered the position most similar to the template feature within this search feature channel. Therefore, the geometric channel $\omega^{i}\in R^{W\times H\times C}$, where *C* is 256, is modeled by calculating the Euclidean distance from the maximum position in each response channel of the regression branch to the remaining feature pixels. Then, the geometric constraints are multiplied element-wise with the corresponding backbone features $F^{i}\in R^{W\times H\times C}$ to obtain saliency features(3)$$S^{i}=\omega^{i}⊙F^{i},i\in \left(3,4,5\right).$$

Algorithm 1 provides a detailed summary of the saliency feature generation process.
**Algorithm 1:** Saliency feature generation
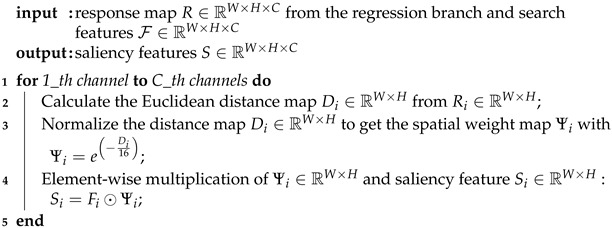


### 3.4. Supervised Training Loss

For fairness, the classification loss $L_{cls}$ and regression loss $L_{reg}$ are kept consistent with baselines. The $L_{cls}$ used in anchor-based or anchor-free trackers is defined as the cross-entropy loss [48]:(4)$$L_{cls}=-\sum_{i}\left(p_{i}\log p_{i}^{*}\right),$$
where $p_{i}$ is the predicted probability of the $i_{th}$ anchor or feature point. $p_{i}^{*}$ is the corresponding classification label. In terms of $L_{reg}$, the anchor-based baseline uses the L1 loss [49], i.e.,(5)$$L_{reg}=\sum_{i}\left|\delta_{i}-\delta_{i}^{*}\right|,$$
where $\delta_{i}$ is the regression prediction and $\delta_{i}^{*}$ is the regression label. In contrast, the anchor-free baseline uses linear loss [50],(6)$$L_{\mathrm{reg}}=\sum_{i}\left(1-IoU\left(\delta_{i},\delta_{i}^{*}\right)\right),$$
where IoU(*) refers to the intersection over union between prediction and ground truth.

Additionally, to obtain high-quality segmentation masks, $L_{\mathrm{mask}\;}$ incorporates the IoU loss [51] and the BCE loss [52], i.e.,(7)$$L_{\mathrm{mask}\;}=\sum_{i=1}^{5}L_{i}^{\mathrm{bce}}+L_{\mathrm{mask}\;}^{iou},$$

Specifically, the loss $L_{i}^{\mathrm{bce}}$ for the mask probability map $m^{\left(i\right)}$ is defined as follows: (8)$$L_{i}^{\mathrm{bce}}=-\sum_{\left(x,y\right)}^{\left(W,H\right)}\left[G_{\left(x,y\right)}\log P_{\left(x,y\right)}+\left(1-P_{\left(x,y\right)}\right)\log\left(1-G_{\left(x,y\right)}\right)\right],$$
where (W,H) is the size of the mask. *P* and *G* are the prediction and label of position (x,y). Meanwhile, the $L_{maz}^{\mathrm{iou}}$ is formulated as: (9)$$L_{maz}^{\mathrm{iou}}=1-\frac{\sum_{x-1}^{H}\sum_{y-1}^{W}P\left(x,y\right)G\left(x,y\right)}{\sum_{x-1}^{H}\sum_{y-1}^{W}\left[P\left(x,y\right)+G\left(x,y\right)-P\left(x,y\right)G\left(x,y\right)\right]},$$

With the above loss function, the overall supervised training loss is expressed as follows:(10)$$L=L_{\mathrm{cls}}+L_{\mathrm{reg}}+L_{\mathrm{mask}\;}.$$

## 4. Experiments

### 4.1. Implementation Details

**Training details**: The experimental hardware platform is an 8-core PC with a 4.5 GHz AMD Ryzen 7000X CPU and a 12 GB Colorful iGame GeForce NVIDIA RTX 4070 GPU. The software platform implemented is the open-source PyTorch 2.1.0. We preserve baseline networks and training schemes. Taking SiamRPN++ [7] and SiamBAN [24] as examples of anchor-based and anchor-free trackers, respectively, we utilize ResNet-50 [53], pre-trained on ImageNet [54], as the backbone, with the first two layers frozen. The classification and regression heads are configured with RPN [7] or BAN [24]. Regarding the training settings, the batch size is 16, and the model is trained for 20 epochs with SGD. During the first 10 epochs, the backbone parameters are frozen and unfrozen in the last 10 epochs with 0.0001 weight decay and 0.9 momentum [24].

**Training datasets**: Our trackers are trained on the training sets of COCO2017 [55], YouTube-VOS [56], and GOT-10k [25]. COCO and YouTube-VOS are widely used datasets for instance segmentation tasks, in which each image has binary mask annotations. For the GOT-10k training dataset that only provides bounding box labels, we employ STA [57] to generate segmentation masks from these boxes. Like SiamMask [21], we crop images of 127×127 and 255×255 pixels as the template and search patches, respectively. Additionally, we employ a series of aggressive augmentations to enhance robustness. This includes random translation (up to ±4 pixels) and scaling (0.95–1.05) of the initial crop; random horizontal flipping (p=0.5); color jittering (p=0.4) via channel-wise offset perturbation; motion blur (p=0.2) with random kernel size and direction; and random grayscale conversion.

**Inference details**: During inference, our trackers produce pixel-wise target masks for all sequences. However, current tracking benchmarks mandate target estimation using a bounding box. For instance, LaSOT [58] asks for axis-aligned boxes, while the VOT2019 [59] requires rotated bounding boxes. Therefore, to evaluate our trackers on these benchmarks, we generate axis-aligned or rotated bounding boxes from masks via the min–max axis-aligned rectangle [21] or the minimum area rectangle [20] methods. All evaluations are conducted with a batch size of 1 to simulate a real-world tracking scenario. During this phase, the incoming test data do not undergo any post-processing, and the reported FPS accounts solely for the time required for data reading and tracker inference.

### 4.2. Quantitative Analysis

We compare the proposed optimization algorithm with cutting-edge trackers on five public tracking benchmarks, i.e., GOT-10k [25], LaSOT [58], VOT2019 [59], UAV123 [60], and OTB100 [61]. Table 1 provides a comprehensive summary of their key characteristics, including the number of video sequences, annotated frames, object classes, attributes per sequence, and specific specializations. The data confirms that the dataset offers extensive coverage across diverse tracking scenarios, such as short-term, long-term, in-the-wild, and UAV-based video tracking. Additionally, Figure 5 visualizes representative image samples from each dataset, illustrating their diversity in content, quality, and appearance.

**GOT-10k**: The GOT-10K testing dataset collects 180 large-scale sequences for general object tracking and uses average overlap (AO) and success rates (SR_0.75_ and SR_0.5_) as indicators. The SOTA and classic trackers included in the comparison are SiamFC [4], SiamRPN [5], SiamFC++-AlexNet [31], SiamMask [21], SiamKPN [35], ATOM [62], SiamCAR [50], and SiamRANK-AlexNet [37], as well as baseline trackers SiamRPN++ [7] and SiamBAN [24]. The results are reported in Table 2. Our optimized trackers, SiamBAN-PMRSL and SiamRPN++-PMRSL, achieve approximately 7.5% and 5.2% higher AO scores compared to the baseline trackers, respectively. SiamBAN-PMRSL outperforms other trackers in all metrics. This validates that our proposed approach effectively improves the tracking performance of Siamese-based trackers and demonstrates strong generalization capabilities.

**LaSOT**: LaSOT is a large-scale tracking benchmark with 280 long-term video sequences, each containing more than 2k frames and annotated with 14 challenging attributes. In this benchmark, we evaluate our algorithms against 12 SOTA trackers using success and precision. As reported in Table 2, SiamBAN-PMRSL and SiamRPN++-PMRSL outperform all trackers in success score, while only trailing behind SiamCorners [34] in precision score. Compared to the baseline trackers SiamBAN [24] and SiamRPN++ [7], our optimized trackers achieve substantial gains (+3.9% and 7.5% in success, respectively, and +2.5% and 7.3% in precision, respectively). Additionally, we further detail the results of our trackers under various challenges, as depicted in Figure 6. Our method effectively improves success and precision compared to the baseline trackers on 14 attributes.

**VOT2019**: VOT2019 has 60 short-term sequences with rotated bounding box annotations. Following the standard VOT evaluation protocol [59], the tracker is reinitialized upon tracking failure. All trackers are evaluated with accuracy (A), robustness (R), and expected average overlap (EAO), which combines both measures. On this dataset, we consider the following Siamese-based trackers: SiamRANK [37], SiamMask [66], SiamRPN [5], SiamMask_E [67], ATOM [62], SRN [68], and the baselines SiamBAN [24] and SiamRPN++ [7]. Figure 7 presents an EAO ranking among these trackers, showing that SiamBAN-PMRSL attains the highest EAO score of 0.324. Compared to the baseline trackers, our SiamBAN-PMRSL and SiamRPN++-PMRSL obtain relative gains of 3.2% and 4.9% in EAO, respectively, which can be attributed to the accurate segmentation and strong discrimination provided by our precise mask representation module and saliency localization head. In terms of accuracy and robustness, our trackers also show improvements, as detailed in Table 2.

**OTB100**: OTB100 is one of the most common tracking benchmarks, which contains 100 fully annotated sequences. Its evaluation criteria are the precision score and the AUC score. The precision score is the percentage of frames in which the distance between the center of tracking results and the ground truth is under 20 pixels [69]. The AUC score is the area under the success plot [61]. On OTB100, we compare our improved algorithm with 12 classic and advanced Siamese-based trackers, i.e., SiamFC [4], SiamRPN [5], SiamRPN++ [7], SiamFC++ [31], SiamBAN [24], Siam R-CNN [17], SiamATL [63], SiamCorners [34], Ocean_Offine [33], SiamGAT [70], SiamAGN [71], and SiamCAR [50]. As reported in Figure 8, our SiamRPN++-PMRSL achieves the best tracking performance, with an AUC score of 0.707 and a precision score of 0.932 (1.5% and 1.8% than baseline, respectively).

**UAV123**: UAV123 is a dedicated unmanned aerial vehicle benchmark, following the OPE [69] protocol. To verify that our proposed strategies can also improve Siamese-based tracker performance on aircraft, we compare our trackers with 10 SOTA Siamese-based trackers; see Figure 9. Our improved SiamBAN++-PMRSL ranks second in both precision and success, while SiamRPN++-PMRSL achieves the highest precision.

### 4.3. Qualitative Analysis

To qualitatively analyze the segmentation tracking accuracy of trackers with the proposed PMR module and SL head, we visualize the results of the improved SiamBAN-PMRSL and SiamRPN++-PMRSL on some challenging sequences from VOT2019 [59], LaSOT [58], and GOT-10k [25].

Figure 10 shows a few qualitative examples of mask predictions for objects of different types and shapes. We observe that our improved trackers perform well with various non-rigid deformation targets (like spider-14, butterfly, bird3, and hand-16) and provide fairly accurate masks even in the presence of similar distractors (e.g., polo, robot1, bolt1, and ants1). In general, our improved trackers not only achieve accurate mask segmentation but also exhibit strong discriminative ability, which is attributed to the collaboration between PMR and SL.

In addition, we present an intuitive comparison of the tracking accuracy between the improved trackers and the original baselines, as shown in Figure 11. The results clearly show that SiamBAN-PMRSL and SiamRPN++-PMRSL achieve better tracking performance than the baseline SiamBAN and SiamRPN++ across multiple attribute scenarios. For instance, in the zebra-17 and 00005 sequences, the improved method successfully tracks the target, while the baseline fails. This improvement can be attributed to the pixel-wise segmentation strategy, which offers a precise representation of the target region and enables clearer identification of its visible parts. In scenarios with significant background clutter, such as the 000094 and 000044 sequences, both baseline SiamRPN++ and SiamBAN struggle to distinguish the target from the background. However, after integrating the proposed PMR and SL modules, the trackers can accurately identify the target. Similarly, in the fernando, motocross1, and airplane-9 sequences, illumination variation often causes baseline trackers to fail, whereas the improved method substantially reduces the impact of such challenging conditions. Collectively, these results demonstrate that the proposed method effectively enhances the robustness and accuracy of the tracker under diverse challenging conditions.

### 4.4. Ablation Study

The precise mask representation (PMR) module is the core component of our algorithm, which decodes multi-scale features to generate pixel-wise target masks for tracking. Additionally, a saliency head that constructs saliency features with spatial constraints is introduced to enhance the discriminatory power of trackers. To expose the contributions of these two components, an ablation study is performed on GOT-10k [25], LaSOT [58], and VOT2019 [59]. The following three kinds of trackers are compared: the baseline trackers (SiamBAN and SiamRPN++), the first improved trackers that only incorporate the PMR module (SiamBAN-PMR and SiamRPN++-PMR), and, finally, the improved trackers that integrate both the PMR module and the SL head (SiamBAN-PMRSL and SiamRPN++-PMRSL). The ablation results are represented in Table 3. It is intuitively evident that both types of improved trackers obtain varying degrees of improvement compared to baseline trackers, where collaborative trackers demonstrate the most significant improvement.

In terms of both speed and model complexity, the more modules added, the higher the computational cost and the slower the operating speed. Figure 12 shows that the PMR and SL modules significantly improve the tracker’s accuracy, albeit with a speed reduction. Notably, the overall operating speed remains consistently high, exceeding 60 FPS—well above the 30 FPS threshold for real-time performance. This demonstrates that our method achieves substantial performance gains while maintaining high efficiency.

In addition, the results show that our SL head is computationally very lightweight. Specifically, compared to SiamBAN-PMR, the SiamBAN-PMR-SL model adds only 0.49 GFLOPs and 0.11 M parameters, corresponding to increases of 0.8% and 0.2%, respectively. Similarly, compared to SiamRPN++-PMR, the SiamRPN++-PMR-SL variant introduces only 0.64 GFLOPs and 0.11 M parameters, which represent increases of just 1% and 0.2%.

Further visual analysis is conducted on GOT-10k_Test_000038 and LaSOT_zebra10, as illustrated in Figure 13. The achievements above stem from several factors. On the one hand, the PMR module functions as a parallel branch, responsible not only for segmenting target masks but also for positively affecting the correlation confidence in the classification and regression branches. On the other hand, the SL head effectively strengthens the robustness of the segmentation network, assisting in obtaining more precise masks even in cases with similar interfering objects.

### 4.5. Failure Examples and Future Work

Based on the experimental results and analysis above, we believe that the proposed segmentation tracking scheme is effective and portable. However, in some extreme scenarios with small-target motion blur and low resolution, our method exhibits tracking failures, as shown in Figure 14. It can be observed that the baseline tracker outperforms the improved trackers in scenarios with motion blur and low resolution. This is primarily due to the following reasons. (1) Under extreme motion blur, details like texture and edges are lost. The mask model must decide between foreground and background for each pixel using unclear cues, resulting in high uncertainty. In contrast, a bounding box offers a coarse, region-level representation. It is more robust to noise because it relies on the general area of the object rather than precise contours, which can often still be detected as a whole even when blurred. (2) In tracking, the prediction from the previous frame is often used as a prior for the next. A noisy mask prediction introduces errors not only in the object’s shape but also in its precise location. These errors are then propagated and amplified in subsequent frames. A bounding box, being a simpler and more stable parameterization (center point, width, and height), is less prone to such accumulation of high-frequency errors. While drift still occurs, its onset might be slower compared to the rapid degradation of a noisy mask.

Therefore, our future work will focus on enhancing the robustness of segmentation tracking from two perspectives: (1) integrating advanced motion deblurring techniques into the tracking pipeline to restore critical image details for the segmentation module and (2) exploring mechanisms to model long-term temporal information (e.g., memory-augmented networks or recurrence-based error correction modules) to detect and mitigate error propagation, thereby improving stability in complex scenarios.

## 5. Conclusions

In this paper, we design a simple and easily portable mask representation plugin, PMRSL, aiming to enable Siamese trackers to achieve precise segmentation tracking instead of bounding box tracking. Our plugin predicts multi-resolution target masks stage by stage through the PMR module while enhancing tracking robustness with an SL head to build saliency features with spatial position constraints. We integrate the proposed models into typical anchor-based and anchor-free Siamese trackers, i.e., SiamRPN++ and SiamBAN. Results on five common tracking datasets demonstrate that our methods can effectively optimize the paradigm of baseline trackers, improving their accuracy and robustness.

## Figures and Tables

**Figure 1 sensors-25-05743-f001:**
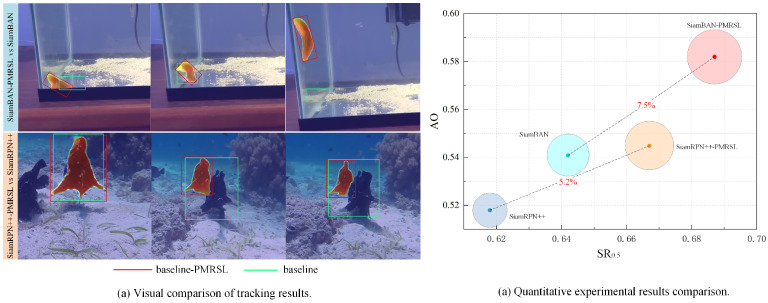
Comparisons on GOT-10k dataset. The left subplot visualizes the tracking results of baseline trackers and their improved versions, denoted as *-PMRSL, integrating the precise mask representation module and saliency localization head. The right subplot represents a quantitative comparison, where AO (Average Overlap) and SR (Success Rate) are used as evaluation metrics. Here, SR_0.5_ denotes the proportion of successfully tracked frames whose overlap ratio exceeds the threshold of 0.5, reflecting the tracking accuracy.

**Figure 2 sensors-25-05743-f002:**
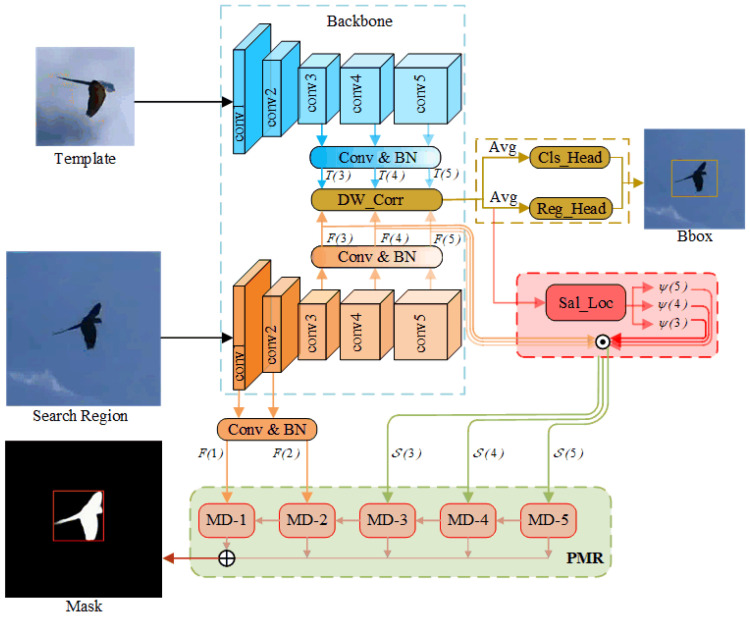
The architecture of our tracking network. It includes four components: the feature extraction backbone, the classification and regression heads, the saliency localization (SL) head, and the precise mask representation (PMR) module. Among them, the first two parts are based on the benchmark Siamese tracker. The SL head constructs saliency with spatial position constraints. The PMR module produces precise target masks.

**Figure 3 sensors-25-05743-f003:**
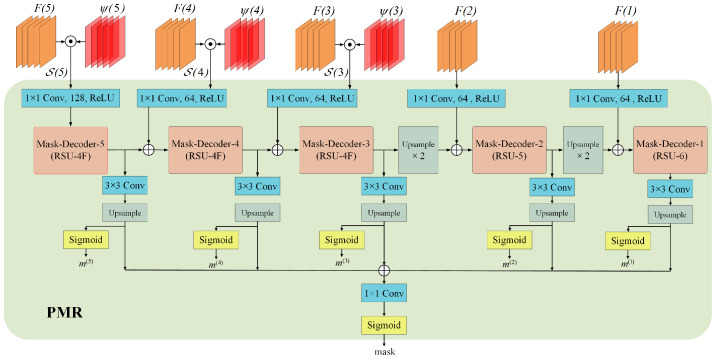
Illustration of our multi-stage precise mask representation module architecture. Each decoder output undergoes a 3×3 convolution followed by upsampling to match the search image size. These outputs are then concatenated and processed through a 1×1 convolution followed by a sigmoid function to predict the target mask. During training, the outputs are processed through a sigmoid function to obtain five probability maps, i.e., $m^{\left(1\right)}$, $m^{\left(2\right)}$, $m^{\left(3\right)}$, $m^{\left(4\right)}$, and $m^{\left(5\right)}$, which, along with the target mask, are involved in loss training to learn a more accurate mask model.

**Figure 4 sensors-25-05743-f004:**
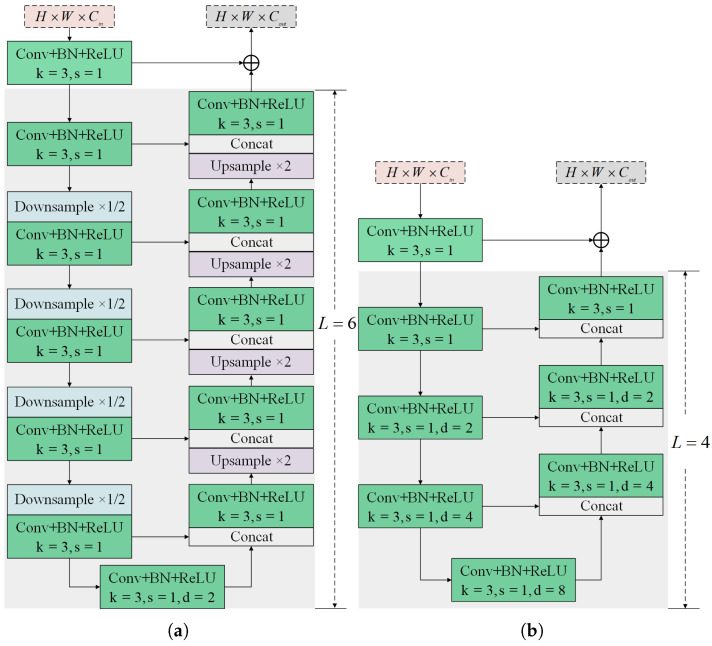
Schematic illustration of our mask decoder with a residual U-block. (**a**) A standard RSU-L with pooling and upsampling. (**b**) A tiny RSU-LF with dilated convolution.

**Figure 5 sensors-25-05743-f005:**
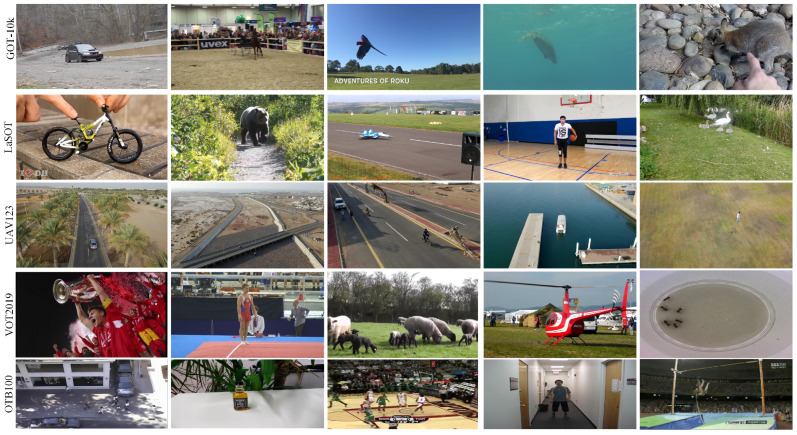
Visualization of representative image samples from each dataset. Five randomly selected examples from each dataset are displayed to illustrate the diversity in content, quality, and appearance.

**Figure 6 sensors-25-05743-f006:**
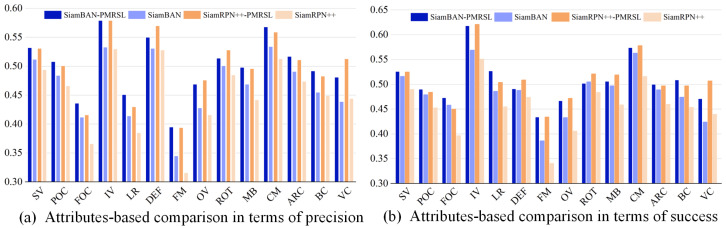
Comparison of precision and success of 14 attributes on LaSOT. Our SiamBAN-PMRSL and SiamRPN++-PMRSL demonstrate substantial improvements in most attributes compared to SiamBAN and SiamRPN++.

**Figure 7 sensors-25-05743-f007:**
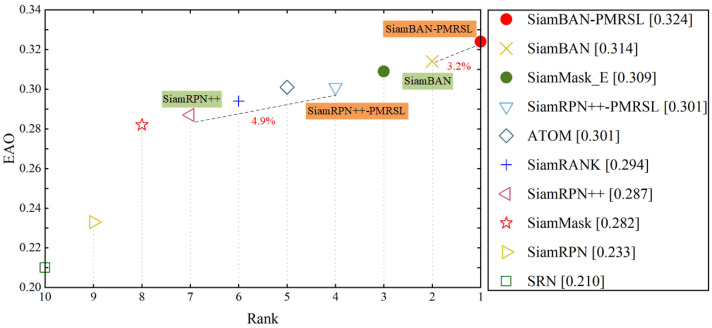
EAO rank plot on VOT2019. Our improved SiamBAN-PMRSL and SiamRPN++-PMRSL obtain relative gains of 3.2% and 4.9% to SiamBAN and SiamRPN++, respectively.

**Figure 8 sensors-25-05743-f008:**
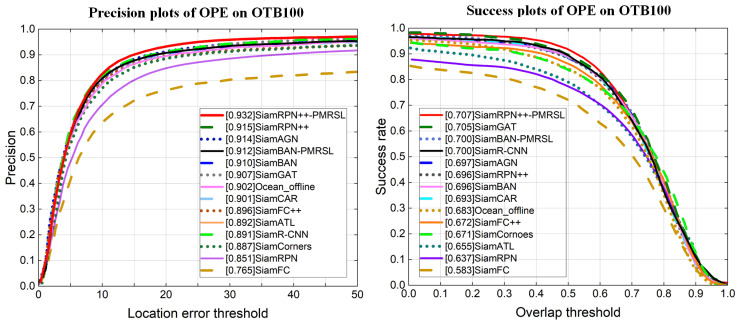
Precision and success plots on OTB100. Our SiamRPN++-PMRSL obtains the top performance.

**Figure 9 sensors-25-05743-f009:**
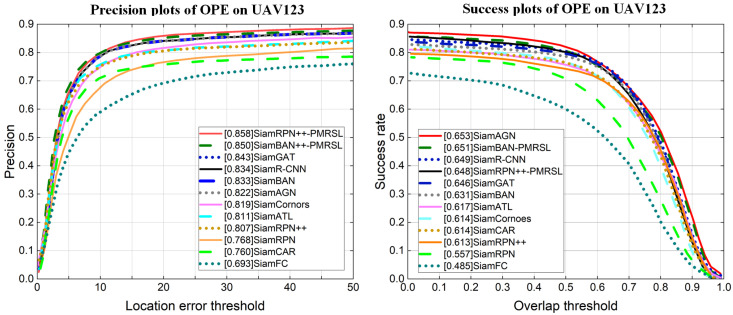
Precision and success plots on UAV123.

**Figure 10 sensors-25-05743-f010:**
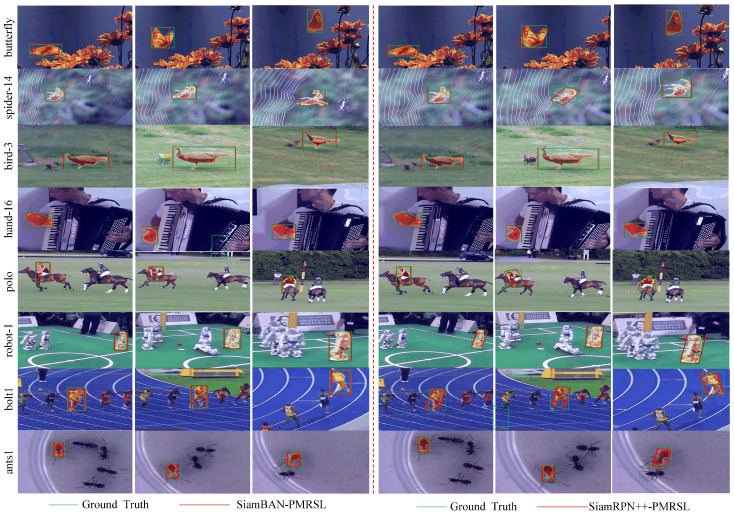
Qualitative results of SiamBAN-PMRSL and SiamBAN-PMRSL for sequences robot-1, spider-14, bird3, and hand-16, polo, butterfly, bolt1, and ants1 from LaSOT and VOT2019. Green and red boxes denote the ground truth and our predictions, respectively. The highlighted red regions represent the mask predictions from our method.

**Figure 11 sensors-25-05743-f011:**
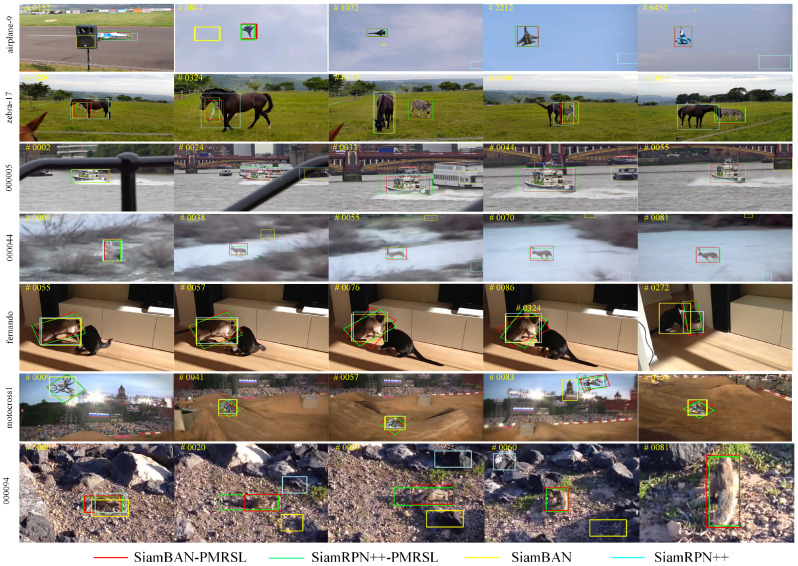
Visualization of the qualitative tracking results of our trackers and the baseline trackers on challenging sequences from LaSOT, GOT-10k, and VOT2019.

**Figure 12 sensors-25-05743-f012:**
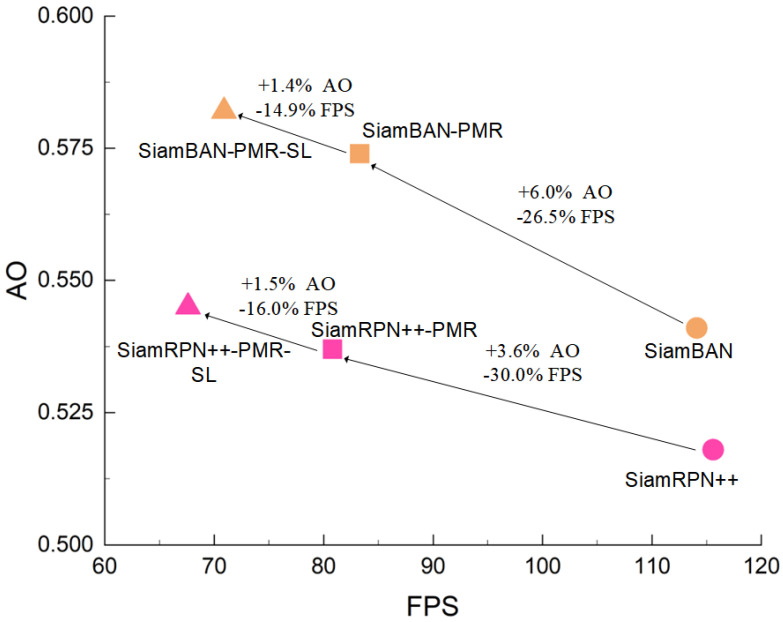
Throughput–accuracy trade-off on the GOT-10k dataset.

**Figure 13 sensors-25-05743-f013:**
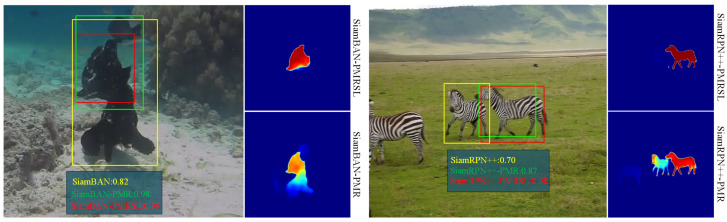
Visualization of tracking results and masks on challenging sequences with background clutter. “*-PMR” or “*-PMRSL” denotes trackers with the PMR module or with both the PMR module and the SL head. Clearly, the tracking confidence scores generated by “*-PMR” and “*-PMRSL” are higher than those of the baseline trackers, while predicting better bounding boxes. In a comparison of masks of “*-PMR” and “*-PMRSL”, the latter can generate more precise target masks, attributable to the discriminative SL head.

**Figure 14 sensors-25-05743-f014:**
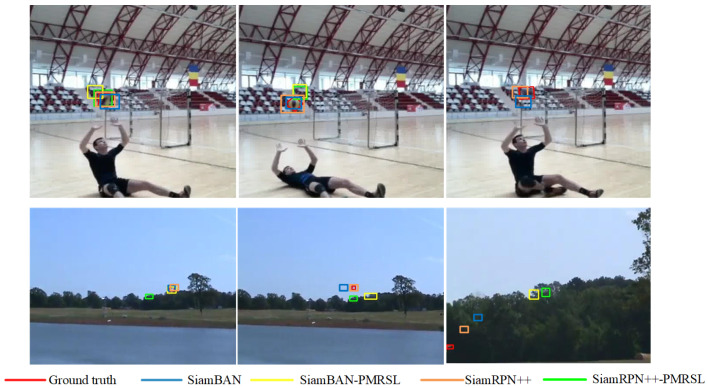
Failure cases: intense motion blur and low-resolution instances.

**Table 1 sensors-25-05743-t001:** Comprehensive comparison of test datasets used in our experiments.

Dataset	No. of Videos	Total Frames	Object Categories	No. of Attributes	Scope
GOT-10k	420	56k	84	6	short-term, wild
LaSOT	280	568k	70	14	long-term, wild
VOT2019	60	19.9k	30	5	short-term, rotated bound Box
UAV123	123	113k	9	12	UAV vision.
OTB100	100	59k	22	11	short-term, general

**Table 2 sensors-25-05743-t002:** Comparison results of cutting-edge trackers on GOT-10k, LaSOT, and VOT2019 datasets. The top three results are highlighted in red, green, and blue.

	GOT-10k			LaSOT		VOT2019		
**Tracker**	**AO**↑	**SR_0.5_**↑	**SR_0.75_**↑	**Succ.**↑	**Prec.**↑	**EAO**↑	**A**↑	**B**↓
SiamFC [4]	0.392	0.426	0.135	-	-	-	-	-
SiamRPN [5]	0.463	0.549	0.253	0.448	0.436	0.233	**0.604**	0.487
SiamFC++-AlexNet [31]	0.493	0.577	0.323	0.501	-	-	-	-
SiamMask [21]	0.514	0.587	0.366	0.467	0.469	0.282	**0.604**	0.487
SiamKPN [35]	0.529	0.606	0.362	0.489	0.489			
ATOM [62]	0.556	0.635	0.402	0.515	-	0.301	0.603	0.411
SiamCAR [50]	0.569	0.687	0.415	0.507	0.510			
SiamRAKN [37]	0.544	0.646	0.368	-	-	0.294	0.588	0.461
SiamATL [63]	0.388	-	-	0.429	0.412	-	-	-
SiamCorners [34]	-	-	-	0.480	**0.555**	-	-	-
Ocean_offline [33]	-	-	-	0.526	0.526	**0.327**	0.590	0.376
SiamDMU [64]	-	-	-	0.499	0.498	-	-	-
ASTABSCF [28]	-	0.581	-	0.460	0.487	-	-	-
MRMACF [65]	-	-	-	0.475	0.500	-	-	-
SiamBAN [24]	0.541	0.642	0.401	0.514	0.518	0.314	0.597	0.381
SiamRPN++ [7]	0.518	0.618	0.325	0.495	0.493	0.287	0.595	0.467
**SiamRPN++-PMRSL**	0.545	0.667	0.317	0.532	0.529	0.301	0.597	0.426
**SiamBAN-PMRSL**	**0.582**	**0.687**	**0.446**	**0.534**	0.531	0.324	0.600	**0.321**

Bold indicates best performance. An upward arrow indicates that a higher value for this metric is better, whereas a downward arrow indicates that a lower value is preferable.

**Table 3 sensors-25-05743-t003:** Ablation study on GOT-10k, LaSOT, and VOT2019.

	PMR	SL	GFLOPs	Params	GOT-10k			LaSOT			VOT2019		
					AO ↑	SR_0.5_ ↑	FPS ↑	Succ. ↑	Prec. ↑	FPS ↑	EAO ↑	A ↑	FPS ↑
SiamBAN			59.59	53.93M	0.541	0.642	**114.1**	0.514	0.518	**127.0**	0.312	0.597	**142.5**
	✓		61.82	54.60M	0.574	0.676	83.3	0.519	0.527	97.2	0.309	0.598	107.6
	✓	✓	62.31	54.71M	**0.582**	**0.687**	70.9	**0.534**	**0.531**	79.4	**0.324**	**0.600**	80.5
SiamRPN++			59.60	53.95M	0.518	0.618	**115.6**	0.496	0.491	**126.5**	0.276	0.603	**142.9**
	✓		61.61	54.55M	0.537	0.641	80.8	0.515	0.521	95.4	0.290	0.609	104.7
	✓	✓	62.25	54.66M	**0.545**	**0.667**	67.6	**0.532**	**0.529**	81.3	**0.301**	**0.597**	87.8

bold indicates best performance.

## Data Availability

Due to privacy or ethical restrictions, data are unavailable.

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
