# Peer review of "Learning Precise Mask Representation for Siamese Visual Tracking"

_sensors, 2025, doi:10.3390/s25185743_

Round 1

Reviewer 1 Report

Comments and Suggestions for Authors

This paper presents a novel and effective plug-and-play module, named PMRSL, designed to enhance existing Siamese visual trackers by incorporating precise mask prediction. The core idea is to supplement the standard bounding box regression with a parallel branch that generates high-quality segmentation masks. The proposed architecture, consisting of a Precise Mask Representation (PMR) module and a Saliency Localization (SL) head, is well-motivated and technically sound. The authors have conducted extensive experiments on six popular benchmarks, demonstrating significant performance improvements over strong baseline trackers (SiamRPN++ and SiamBAN) while maintaining real-time speeds. The paper is well-structured, and the contributions are clearly articulated. I recommend this paper for acceptance after some minor revisions.

Comments

1.Clarification of the Saliency Localization (SL) Head: The description of the SL head in Section 3.3 is concise but could benefit from more detail. The paper states it computes the Euclidean distance from the maximum point in the response map to all other pixels. It would be helpful to explicitly describe:

How is this distance map normalized or scaled before being multiplied with the search features? Is this operation performed independently for each of the 256 channels, or is a single spatial weight map generated and applied to all channels? A more detailed mathematical formulation or a small pseudo-code block would enhance clarity and reproducibility.

2.Expanded Ablation Study: The ablation study in Table 2 is very informative but could be slightly expanded to be more conclusive.

It currently compares baseline, baseline+PMR, and baseline+PMR+SL. An interesting addition would be to analyze the effect of the SL head alone on the baseline's bounding box prediction. Does providing this spatial saliency feature back to the classification/regression heads improve their performance, even without the mask branch?

While FPS is a useful metric, reporting the computational cost in GFLOPs for the baseline and the additional components (PMR, SL) would provide a more hardware-independent measure of the added complexity.

3.Deeper Analysis of Failure Cases: Section 4.5 rightly points out that the method can fail in scenarios with extreme motion blur and low resolution, attributing this to a lack of temporal information. The analysis could be slightly deepened by discussing why a mask-based representation might be more vulnerable in these cases compared to a bounding box. For instance, does the pixel-level prediction become overly noisy and unreliable, leading to faster model drift? A brief discussion on this would strengthen the paper.

4.In the caption for the ablation study table, the reference is "Table ??". This should be corrected to "Table 2".

5.In Figure 11, the text within the images (e.g., "SiamBAN-0.82") is slightly difficult to read. Improving the resolution or using a clearer font would be helpful.

Reviewer 2 Report

Comments and Suggestions for Authors

This paper presents a novel segmentation-assisted model for binary mask representation of targets, enabling pixel-level segmentation tracking within various Siamese frameworks. The approach integrates (i) a multi-stage precise mask representation module with cascaded U-Net decoders for accurate mask prediction, and (ii) a saliency localization head employing the Euclidean model to impose spatial constraints and enhance decoder discrimination. Experiments across six tracking benchmarks confirm significant performance gains for both anchor-based and anchor-free Siamese trackers. I recommend some adjustments to further improve the presentation of the results and to remove some ambiguity before recommending this paper for acceptance:

  1. The authors claim to have conducted experiments on six tracking benchmarks in the manuscript; however, only five dataset evaluations are presented.
  2. The dataset used in this method need more information and give some sample image data.
  3. The quantitative and qualitative results are not enough. The authors should include a separate section with resulting images to make the readers easy understanding.
  4. How to balance the computational cost and tracking accuracy improvement in this method?
  5. There are some issues with writing in various parts of the article. Hence, the writing of this paper needs further polishing. For example “while SiamRPN++-PMRSL gets the highest pre” in UAV123.
  6. I suggest that the authors discuss the limitations of the proposed method in detail Section 4.5. In fact, any method has its own limitations and shortcomings. Also, add more discussions about the future directions of this research.
  7. Overall, the literature review should be improved. Without discussing the previous works, the novelty of the paper cannot be assessed properly. Discuss the more new papers in related work section such as https://doi.org/10.1016/j.image.2025.117305.

Reviewer 3 Report

Comments and Suggestions for Authors

Overall Evaluation
This manuscript proposes an improved visual object tracking framework that integrates a PMR module and an SL head, validated on multiple benchmark datasets. The overall methodology is clearly presented, and experimental results demonstrate accuracy gains while maintaining high speed, indicating practical and engineering value.However, there is still room for improvement in terms of writing quality, formula definitions, experimental details, and consistency of expressions. 
1. Writing and Formatting
Terminology and spelling consistency: In Section 3.1, “demote” is likely a typographical error for “denote”. In Figure 6, “PMRS” should be unified with “PMRSL” used elsewhere in the manuscript. The method name appears as “SiamCornors / SiamCornor / SiamCorners” in different places; please standardize to the correct spelling.
Software version accuracy: “PyTorch 2.10” is most likely “PyTorch 2.1.0” and should be corrected.
2. Formulas and Notation
Completeness of formula definitions: In Equation (1), the operator is undefined and written in a non-standard form. Consider using the conventional element-wise multiplication symbol ⊙\odot⊙ and explicitly defining its meaning.
Notation and subscript consistency: Some formulas contain inconsistent symbol formatting (e.g., spacing in subscripts, bracket usage). A uniform mathematical notation style throughout the manuscript will improve clarity and rigor.
3. Logical Consistency
Table presentation: Table 1 currently uses color to highlight performance rankings, which may be ineffective in grayscale or printed formats. Consider using boldface or markers (▲/◆/●) in the cells and explain the marking scheme in the caption.
Dataset count consistency: Both the abstract and Section 4.2 state that experiments were conducted on “six” benchmarks, yet only five datasets (GOT-10k, LaSOT, VOT2019, UAV123, OTB100) are listed. Please add the missing dataset or adjust the description to “five.”
4. Experiments and Result Presentation
FPS–accuracy trade-off: The FPS varies notably across baselines and datasets. Consider plotting a throughput–accuracy curve (AUC/EAO vs. FPS) under the same hardware/environment to better illustrate the performance–efficiency balance of PMR and SL.
Readability of qualitative results: In Figures 9–11, please indicate the meaning of each color (e.g., GT box, predicted box, mask) and add zoomed-in views of key regions to emphasize improvements.
Currency of comparison methods: Some baseline trackers in Table 1 are outdated. Replacing or supplementing them with more recent representative methods would improve the relevance and impact of the experimental comparison.
Reproducibility: Please include more details about training data augmentation strategies, evaluation settings (e.g., batch size, whether post-processing is included in FPS), and the quality assessment of pseudo masks (e.g., IoU statistics).

Reviewer 4 Report

Comments and Suggestions for Authors

1. In Figure 1, on the right side, what are the values on the x and y axes? What are their units? The authors should briefly explain, either in the text before or after the figure, or in the figure caption, what AO and SR0.5 are, as is done for other evaluation metrics in subsequent figures and tables. Alternatively, a reference can be provided as well. While these parameters are popular among field experts, the journal is open access, and the average reader may not be familiar with them.

2. Table 2 is referenced in the text as Table?? (page 13). This needs to be corrected.

3. In Table 2, shouldn’t the largest FPS values be in bold? Below the table, it is indicated what bold is used for, but lower values of FPS are linked with worse performance. Alternatively, FPS could mean something else, but the authors should clearly explain that.

4. Can the authors provide more details on the exact AIB of the GPU used? (e.g. ASUS, Palit, etc.) This is important because different manufacturers may overclock graphics cards, potentially leading to different results if another research team tries to replicate the study's findings.

5. It would be more appropriate to put the baseline SiamRPN, SiamRPN++ and SiamRPN++-PMRSL next to each other in Table 1 in order to show the increase in performance due to the application of the method. The same logic can be applied to the pair SiamBAN and SiamBAN-PMRSL. It would increase readability and highlight the usefulness of the method.

6. The last sentence on page 11, before Figure 8, stops abruptly: “Our improved SiamBAN++-PMRSL ranks second in both precision and success, while SiamRPN++-PMRSL gets the highest pre…”. This needs to be corrected.

7. Why are there five mask decoders in the PMR module? Can the method perform well with a different number of mask decoders?

Round 2

Reviewer 2 Report

Comments and Suggestions for Authors

The authors have addressed all my review comments. The current version can be accepted for publication.

Reviewer 3 Report

Comments and Suggestions for Authors

I have no more comments